# Association of visceral adiposity index and lipid accumulation products with prediabetes in US adults from NHANES 2007–2020: A cross-sectional study

**Li-Ting Qiu, Ji-Dong Zhang, Bo-Yan Fan, Ling Li, Gui-Xiang Sun** *

The College of Traditional Chinese Medicine, Hunan University of Chinese Medicine, Changsha, P. R. China

* 84663423@qq.com

## Abstract

### Background

The lipid accumulation product (LAP) and the visceral adiposity index (VAI) are suggested as dependable measures for assessing visceral fat levels. Prediabetes is recognized as a condition that precedes the potential onset of diabetes. The objective of this research is to investigate how VAI and LAP are related to prediabetes among the adult population in the United States.

### Methods

Information from the 2007–2020 National Health and Nutrition Examination Survey (NHANES) was scrutinized in a cross-sectional study. To evaluate the connection between VAI or LAP and the presence of prediabetes, both univariate analysis and multivariate logistic regression were utilized. Threshold effect analysis and fitted smoothing curves were used to delve into the non-linear association between VAI or LAP and prediabetes. Additional analyses were performed on specific subgroups, along with tests to explore potential interactions.

### Results

In general, 12,564 American adults were included. After full adjustment, prediabetes with VAI (OR: 1.128, 95% CI: 1.073–1.185) or LAP (OR: 1.006, 95% CI: 1.004–1.008) showed a positive correlation. Individuals in the 4th VAI quartile group faced a significant 61.9% elevated risk for prediabetes (OR: 1.619, 95% CI: 1.354–1.937) when contrasted to those in the 1st VAI quartile. Participants in the 4th LAP quartile group had a significant 116.4% elevated risk for prediabetes (OR: 2.164, 95% CI: 1.747–2.681) when contrasted to individuals of the 1st LAP quartile. Smooth curve fitting analysis revealed a nonlinear correlation of VAI or LAP and prediabetes, and threshold effect analysis was used to determine an inflection point of 4.090 for VAI and 68.168 for LAP.

**Data Availability Statement:** The data for this study are already publicly available through the National Center for Health Statistics (NCHS), National Health and Nutrition Examination Survey

(NHANES) website: https://www.cdc.gov/nchs/nhanes/about_nhanes.htm.

**Funding:** This study was supportded by the National Natural Science Foundation of China (Grant No.81973670), Central Subsidized Chinese Medicine Special Funds - National Medical Master Sun Guangrong Hunan Workshop Project (2015). The funders had no role in study conceptualization, data curation, formal analysis, methodology, software, decision to publish, or preparation of the manuscript. There was no additional external funding received for this study.

**Competing interests:** The authors have declared that no competing interests exist.

**Abbreviations:** BMI, body mass index; CI, confidence interval; CVD, cardiovascular diseases; HDL, high density lipoprotein-cholesterol; LAP, lipid accumulation product; METs, metabolic equivalent scores; NHANES, National Health and Nutrition Examination Survey; OR, odds ratio; PIR, poverty income ratio; TG, triglyceride; VAI, visceral obesity index; WC, waist circumference; WHR, waist to hip ratio.

## Conclusions

The values of VAI and LAP are positively associated with the prevalence of prediabetes. The VAI and LAP indices may be used as predictors of prediabetes.

## Introduction

Type 2 diabetes mellitus (diabetes) is a progressive metabolic disease that poses a significant risk to public health. Timely identification of high-risk individuals could help prevent and control diabetes. Prediabetes is a well-acknowledged risk factor for future diabetes, representing an intermediate state where plasma glucose levels range between normal glucose metabolism and diabetes [1, 2]. Research evidence indicates a link between prediabetes and cardiovascular disease (CVD), diabetic retinopathy, neuropathies, and kidney disease [3–6]. Identifying and addressing prediabetes at an early stage can successfully halt the progression to diabetes and its associated health issues [7]. Studies indicate that each year, approximately 5–10% of individuals with prediabetes develop diabetes [2], and the conversion rate can be up to 70% [8]. The International Diabetes Federation projects that by 2030, close to 470 million individuals will be affected by prediabetes. The challenge in diagnosing prediabetes lies in its non-specific symptoms, making it a condition that often goes undetected. Therefore, awareness of the prediabetes risk factors is essential, as timely and appropriate intervention may reverse the incidence of diabetes and related complications.

A substantial body of research indicates that individuals carrying excess weight or with obesity are at an increased risk for the onset of prediabetes [9, 10]. Particularly, individuals with centripetal distribution of adiposity (visceral adiposity) are believed to have a greater predisposition to prediabetes compared to individuals with subcutaneous adiposity [11, 12]. Techniques like magnetic resonance imaging (MRI) and computed tomography (CT) are accurate in assessing the distribution of body fat, but their high costs and practical limitations curtail their widespread application in both research and routine clinical settings. Traditional obesity-related indicators, such as body mass index (BMI) and waist circumference (WC), are non-invasive and readily available but fall short in accurately distinguishing between subcutaneous and visceral fat masses. The visceral adiposity index (VAI), which integrates both anthropometric measurements (including BMI and WC) and blood biomarkers (including triglyceride (TG) and high density lipoprotein-cholesterol (HDL)), has gained acceptance as an effective measure for assessing the function of visceral fat [13]. Accordingly, the VAI is particularly adept at detecting metabolically unhealthy profiles that are often linked with central fat accumulation, including cardiovascular disease, metabolic syndrome, and insulin resistance [14–17]. Lipid accumulation product (LAP) is an index integrates WC and TG and is a measure of abdominal lipid accumulation [18]. LAP indicators are effective at identifying insulin sensitivity, diabetes, metabolic syndrome, and CVD when contrasted to traditional lipid profiles [19–22]. There are several studies analyzing the relationship between VAI or LAP and prediabetes. In a study of the Montenegrin population conducted by Klisic et al. [23] a significant positive relationship with prediabetes was suggested for both VAI and LAP. Ramdas Nayak et al. [24] found that both VAI and LAP were generally more effective than WC, waist to hip ratio (WHR), and BMI at predicting prediabetes within the Asian Indian demographic. Ahn et al. [25] demonstrated the efficacy of VAI and LAP as valuable indicators for discerning prediabetes/diabetes within a German population. Nevertheless, the ethnic specificity of visceral fat mean their results may not be widely applicable. Moreover, there has been a lack of

investigation into the connection between VAI, LAP, and prediabetes across a nationally representative cohort of adults in the United States.

Therefore, we explored the correlation between prediabetes and VAI or LAP in a larger and more representative sample of various ethnic groups in the United States. The hypothesize of this study is that lower levels of VAI or LAP are associated with a significantly lower risk of prediabetes.

## Methods

### Study design and participants

14 years of data (2007 to 2020) from the National Health and Nutrition Examination Survey (NHANES) were applied. The NHANES, a database compiled by the Centers for Disease Control (CDC) and Prevention's National Center for Health Statistics (NCHS), is constructed from a population-based national survey for the evaluation of the noninstitutionalized American population's nutritional status and health. Participants were recruited by a complex and multistage probability sampling design that continuously samples approximately 5000 participants annually in the United States to ensure the representation of diverse demographic groups [26]. Specifically, NHANES first selects primary sampling units (PSUs) across the country, followed by stratified random sampling within each PSU to choose households, and finally, individuals are randomly selected from eligible household members. Survey participants were requested to partake in a household interview, encompassing questionnaires regarding socio-demographic, dietary, and general health information, along with a medical examination conducted at mobile examination centers, encompassing medical, dental, and physiological measurements. Initially, 66,148 people were involved in the study from 2007 to 2020 in the NHANES database. The study did not include the following groups: (1) individuals below the age of 20; (2) participants with diabetes data; (3) participants without information on diabetes, prediabetes, and blood glucose levels; (4) those without available information regarding VAI and LAP. After applying these criteria, the study population comprised 12,564 subjects (Fig 1). This investigation adhered to the ethical principles outlined in the Declaration of Helsinki [27]. Written consent was obtained from all participants of the NHANES study, and the project received ethical clearance from the NCHS Research Ethics Review Board [28].

### Exposure variable and outcomes

Individuals were identified as having prediabetes based on one or more of the following criteria: an HbA1c range of 5.7 to 6.4%, an impaired fasting glucose range of 5.6 to 7.0 mmol/L, an impaired glucose tolerance range of 7.8 to 11.1 mmol/L, or having received a clinical diagnosis of prediabetes from a healthcare provider [29]. The calculations for VAI and LAP were performed using established equations [13, 18].For men, the VAI formula is [13]: $[WC/[(39.68 + (1.88 \times BMI))] \times (TG/1.03) \times (1.31/HDL)]$; for women, it is: $[WC/(36.58 + (1.89 \times BMI))] \times (TG/0.81) \times (1.52/HDL)$. The LAP formula for men is [18] $(WC-65) \times TG$; for women, it is: $(WC-58) \times TG$. Within these formulas, WC is expressed in cm and BMI in kg/m$^2$. TG and HDL cholesterol levels are measured in mmol/L.

### Covariates

Based on previous related studies [30, 31], we evaluated the potential risk factors for prediabetes, which included sociodemographic data (age, sex, race, education, marital, poverty income ratio (PIR)), lifestyle behavior characteristics (smoking status, alcohol use, BMI, physical activity, daily energy intake), and disease history (hyperlipidemia, hypertension, cancer, CVD). The

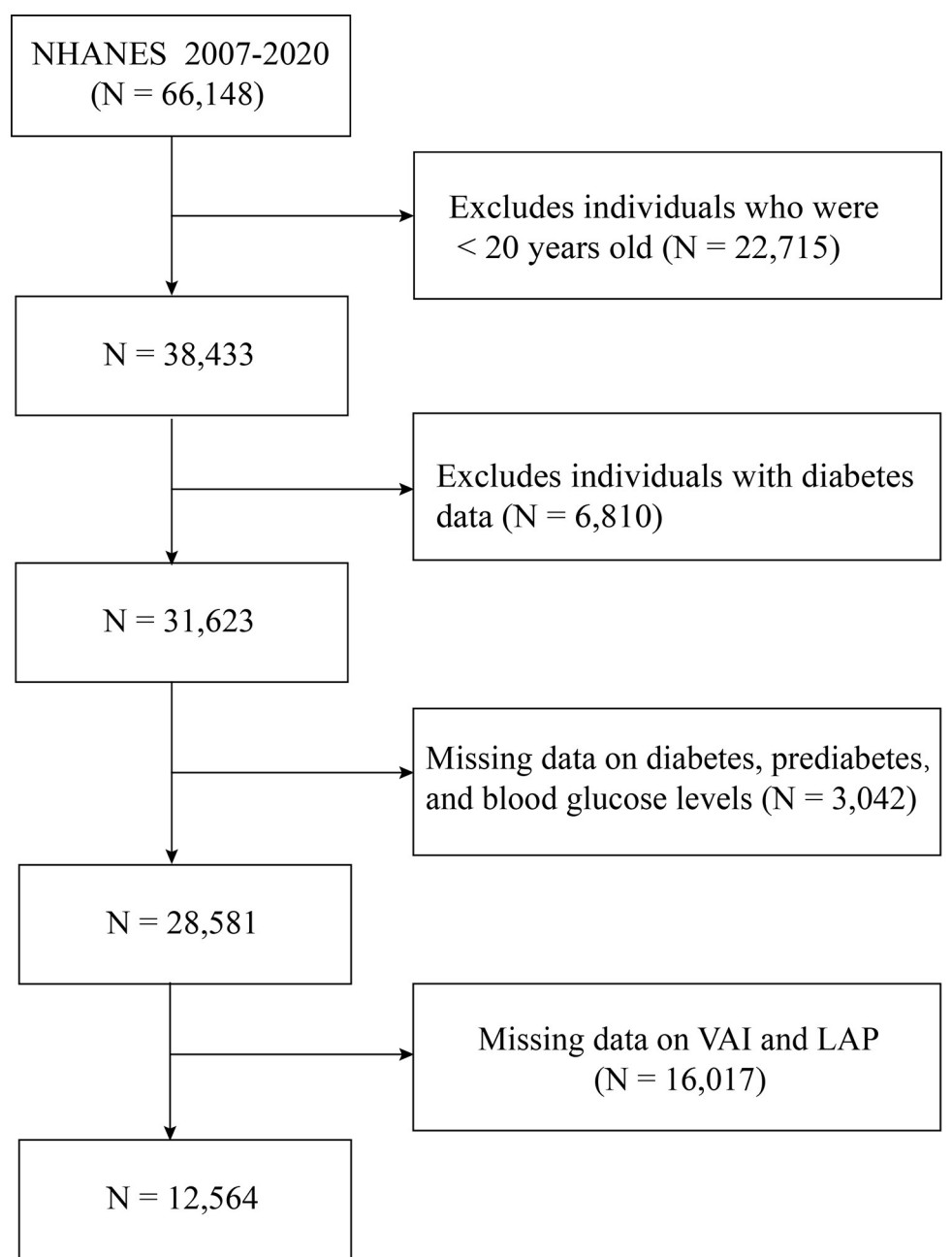

**Fig 1. Flowchart of participant selection.** Abbreviation: NHANES, National Health and Nutrition Examination Survey.

smoking habits of participants were classified into three distinct types: never, former, and current smoker. Participants' alcohol consumption was assessed by the single choice questionnaire, "In the past 12 months, on those days that you drank alcoholic beverages, on the average, how many drinks did you have?" Physical activity was assessed by metabolic equivalent scores (METs) = sum of walking + moderate + vigorous MET-minutes/week scores [32]. The total daily energy intake (kcal) was calculated from the first day of 24-h dietary recall. Hyperlipidemia was characterized by either the use of medication to lower lipid levels or by

meeting any of the following criteria: total cholesterol being equal to or more than 200 mg/dL, TG being equal to or more than 150 mg/dL, low-density lipoprotein levels being equal to or more than 130 mg/dL, or HDL cholesterol being equal to or less than 50 mg/dL for females and being equal to or less than 40 mg/dL for males [33]. Hypertension was identified in individuals with a systolic blood pressure of 140 mmHg or higher and/or a diastolic blood pressure of 90 mmHg or higher, those who reported a history of high blood pressure, or those taking medication to manage blood pressure. The presence of cancer was determined by a positive response to the question, "Have you ever been told you had cancer or a malignancy?" CVD was classified as having been diagnosed with conditions such as stroke, heart attack, angina, coronary heart disease, or coronary heart disease by a healthcare staff.

## Statistical analysis

EmpowerStates (www.empowerstats.com) was used for all statistical analyses. SDMVSTRA and SDMVPSU were utilized to ensure accurate national estimates due to the complex survey design employed in NHANES. All continuous variables had the expression of mean values (95% CI) during the baseline analysis, while categorical variables had the expression of percentages (95% CI). Multiple imputation was conducted to adequately account for missing covariates. The univariate analysis was used to investigate the potential correlation of each covariate and prediabetes. Multiple regression analysis was conducted using three models with different adjustments for confounders: Model 1 was unadjusted, Model 2 adjusted for gender, age, and race, and covariates were retained in the full model 3 if the change in the effect estimate exceeded 10%. Subgroup analyses were conduced by age, gender, smoking status, alcohol use, BMI, physical activity, daily energy intake, hyperlipidemia, hypertension, cancer and CVD. Interaction *P*-values were used to evaluate the consistency of effects across subgroups. We carried out smooth curve fitting for discovering underlying nonlinear relationships. A threshold effect analysis was further performed to demonstrate the association and inflection point between VAI, LAP, and prediabetes. A *P*-value < 0.05 exhibited significance.

## Results

### Participant characteristics

Among the 12,564 participants, there were 7,051 with prediabetes and 5,513 with normal blood glucose. When contrasted to participants with normal blood glucose, those with prediabetes were more likely to be older (50.4, *P* < 0.0001); more frequently male (52.8%, *P* < 0.0001); have a lower proportion of some college and college graduates or above (*P* < 0.0001); be less often never married (14.3%, *P* < 0.0001); be less frequently non-smokers (52.0%, *P* = 0.0007); have a normal weight less often (23.5%, *P* < 0.0001); engage in physical activity < 600 MET-minutes/week more frequently (46.3%, *P* < 0.0001); and have higher VAI (2.1, *P* < 0.0001) and LAP (60.6, *P* < 0.0001); they also had a higher risk of hyperlipidemia (78.1%, *P* < 0.0001), hypertension (42.2%, *P* < 0.0001), cancer (10.9%, *P* < 0.0001), and CVD (9.1%, *P* < 0.0001) (Table 1).

### Univariate analysis

With the aim of examining the correlation of variables and prediabetes, we carried out univariate analysis. Age was found to be positively associated with prediabetes (OR: 1.04, 95% CI: 1.04–1.05). When contrasted to men, women had a lower risk of developing prediabetes (OR: 0.63, 95% CI: 0.59–0.68). Among ethnic groups, non-Hispanic whites (OR: 0.80; 95% CI: 0.72–0.89) and other races (OR: 0.83, 95% CI: 0.72–0.95) showed a lower prevalence of prediabetes.

**Table 1. Characteristics of participants by prediabetes status, NHANES 2007–2020.**

| Variable | Normal blood glucose (N = 5,513) | Prediabetes (N = 7,051) | *P*-value |
|---|---|---|---|
| Age, years | 40.2 (39.5, 40.9) | 50.4 (49.9, 51.0) | <0.0001 |
| Sex | | | <0.0001 |
| Men | 42.4 (40.9, 43.9) | 52.8 (51.3, 54.3) | |
| Women | 57.6 (56.1, 59.1) | 47.2 (45.7, 48.7) | |
| Race and ethnicity | | | 0.1638 |
| Mexican American | 7.9 (6.7, 9.4) | 9.0 (7.5, 10.7) | |
| other Hispanic | 6.2 (5.1, 7.6) | 6.1 (5.1, 7.3) | |
| non-Hispanic white | 68.5 (65.6, 71.2) | 66.8 (64.0, 69.5) | |
| non-Hispanic black | 9.7 (8.4, 11.1) | 10.1 (8.8, 11.5) | |
| other races | 7.7 (6.7, 8.8) | 8.1 (7.1, 9.1) | |
| Education | | | <0.0001 |
| < 9th grade | 3.4 (2.9, 4.1) | 5.9 (5.2, 6.8) | |
| 9–11th grade | 9.1 (7.9, 10.6) | 11.1 (10.1, 12.2) | |
| high school graduate | 19.4 (17.8, 21.0) | 24.3 (22.6, 26.0) | |
| some college | 32.2 (30.2, 34.1) | 29.6 (28.0, 31.3) | |
| college graduate or above | 35.9 (33.4, 38.5) | 29.1 (26.9, 31.4) | |
| Marital | | | <0.0001 |
| Married/Living with partner | 61.9 (59.9, 63.8) | 66.2 (64.1, 68.1) | |
| Widowed/Divorced/Separated | 13.5 (12.4, 14.8) | 19.5 (18.1, 21.1) | |
| Never married | 24.6 (22.6, 26.6) | 14.3 (13.0, 15.7) | |
| Smoking status | | | <0.0001 |
| Never | 61.0 (58.7, 63.3) | 52.0 (50.0, 53.9) | |
| Former | 19.8 (18.3, 21.4) | 27.6 (25.9, 29.4) | |
| Now | 19.2 (17.4, 21.1) | 20.4 (19.0, 21.9) | |
| Alcohol use | | | 0.3845 |
| 1–14 | 99.3 (98.9, 99.5) | 99.4 (99.2, 99.6) | |
| ≥ 15 | 0.7 (0.5, 1.1) | 0.6 (0.4, 0.8) | |
| PIR | | | 0.1848 |
| ≤ 1.3 | 22.3 (20.5, 24.2) | 21.7 (20.1, 23.4) | |
| > 1.3 and ≤ 3.5 | 34.1 (31.9, 36.2) | 36.1 (34.4, 37.9) | |
| > 3.5 | 43.6 (41.1, 46.3) | 42.2 (39.8, 44.5) | |
| BMI | | | <0.0001 |
| Normal (<25 kg/m$^2$) | 43.0 (41.0, 44.9) | 23.5 (22.0, 25.0) | |
| Overweight (25–30 kg/m$^2$) | 33.0 (31.4, 34.7) | 35.4 (34.0, 36.9) | |
| Obese (≥30 kg/m$^2$) | 24.0 (22.4, 25.7) | 41.1 (39.4, 42.8) | |
| Physical activity | | | 0.0007 |
| < 600 | 43.4 (42.0, 44.9) | 46.3 (44.7, 47.9) | |
| 600–1500 | 31.8 (30.3, 33.3) | 28.0 (26.7, 29.4) | |
| > 1500 | 24.8 (23.4, 26.3) | 25.6 (24.1, 27.2) | |
| Daily energy intake (kcal) | 2196.3 (2161.6, 2231.0) | 2204.4 (2169.4, 2239.4) | 0.7555 |
| VAI | 1.5 (1.4, 1.5) | 2.1 (2.0, 2.1) | <0.0001 |
| LAP | 38.8 (37.4, 40.2) | 60.6 (58.6, 62.6) | <0.0001 |
| Hyperlipidemia | | | <0.0001 |
| No | 41.8 (40.0, 43.6) | 21.9 (20.6, 23.4) | |
| Yes | 58.2 (56.4, 60.0) | 78.1 (76.6, 79.4) | |
| Hypertension | | | <0.0001 |
| No | 79.3 (77.5, 81.0) | 57.8 (56.0, 59.5) | |

(*Continued*)

**Table 1.** (Continued)

| Variable | Normal blood glucose (N = 5,513) | Prediabetes (N = 7,051) | *P*-value |
|---|---|---|---|
| Yes | 20.7 (19.0, 22.5) | 42.2 (40.5, 44.0) | |
| Cancer | | | <0.0001 |
| No | 93.5 (92.6, 94.4) | 89.1 (88.0, 90.2) | |
| Yes | 6.5 (5.6, 7.4) | 10.9 (9.8, 12.0) | |
| CVD | | | <0.0001 |
| No | 96.6 (96.0, 97.2) | 90.9 (89.9, 91.9) | |
| Yes | 3.4 (2.8, 4.0) | 9.1 (8.1, 10.1) | |

Continuous variables were listed as weighted mean (95% CI). Categorical variables were listed as weighted percentage (95% CI). Abbreviations: BMI, body mass index; CVD, cardiovascular diseases; LAP, lipid accumulation product; NHANES, National Health and Nutrition Examination Survey; PIR, poverty income ratio; VAI, visceral obesity index.

Regarding education level, individuals with 9–11th grade education (OR: 0.67, 95% CI: 0.57–0.79), high school graduates (OR: 0.65, 95% CI: 0.56–0.76), some college education (OR: 0.51, 95% CI: 0.44–0.59), and college graduates or above (OR: 0.46, 95% CI: 0.39–0.53) had lower prevalence rates of prediabetes when contrasted to individuals with less than a 9th-grade education. When contrasted to being married/living with a partner, the widowed/divorced/separated group (OR: 1.34, 95% CI: 1.22–1.47) had a higher risk of prediabetes. Those who were never married (OR: 0.54, 95% CI: 0.49–0.59) had a lower prevalence of prediabetes. Former smokers (OR: 1.64, 95% CI: 1.50–1.79) and current smokers (OR: 1.16, 95% CI: 1.06–1.27) were at a higher risk of prediabetes than nonsmokers. Obesity (OR: 2.79, 95% CI: 2.56–3.06) and being overweight (OR: 1.87, 95% CI: 1.71–2.04) significantly elevated the odds of prediabetes when contrasted to normal weight. Engaging in physical activity of 600–1500 MET-minutes/week (OR: 0.84, 95% CI: 0.77–0.93) was associated with a lower risk of prediabetes than physical activity below 600 MET-minutes/week. Notably, VAI (OR: 1.26, 95% CI: 1.22–1.29) and LAP (OR: 1.01, 95% CI: 1.01–1.01) showed a positive correlation to the occurrence of prediabetes. Additionally, hyperlipidemia (OR: 2.35, 95% CI: 2.18–2.54), hypertension (OR: 2.79, 95% CI: 2.58–3.02), cancer (OR: 1.86, 95% CI: 1.62–2.14), and CVD (OR: 2.50, 95% CI: 2.15–2.91) were all positively related to prediabetes (Table 2).

## Multivariate regression analysis

Three multiple regression models were applied to explain the correlation between prediabetes and VAI and LAP. The VAI and LAP levels were stratified into quartiles, with the lowest quartile (Q1) serving as the reference. The results indicated a significant positive correlation between VAI and prediabetes in models 1 (OR: 1.294, 95% CI: 1.230–1.361), 2 (OR: 1.293, 95% CI: 1.226–1.363), and 3 (OR: 1.128, 95% CI: 1.073–1.185). When VAI was categorized into quartiles, participants in the highest quartile (Q4: ≥ 2.13) showed a positive correlation to the prevalence of prediabetes across models 1 (OR: 2.885, 95% CI: 2.520–3.302), 2 (OR: 2.912, 95% CI: 2.504–3.386), and 3 (OR: 1.619, 95% CI: 1.354–1.937). All *P* for trend < 0.0001 (Table 3).

Prediabetes was significantly correlated with LAP in models 1 (OR: 1.014, 95% CI: 1.012–1.016), 2 (OR: 1.012, 95% CI: 1.011–1.014), and 3 (OR: 1.006, 95% CI: 1.004–1.008). Participants in the highest LAP quartile (Q4: ≥ 62.28) showed a substantially higher risk of developing prediabetes in models 1 (OR: 5.236, 95% CI: 4.499–6.094), 2 (OR: 4.286, 95% CI: 3.649–5.036), and 3 (OR: 2.164, 95% CI: 1.747–2.681) when contrasted to individuals in the lowest quartile (Q1: < 20.41). The *P* for trend was < 0.0001 across all models (Table 3).

**Table 2. Association of unadjusted variables with prediabetes.**

| Variables | OR (95%CI) | *P*-value |
|---|---|---|
| Age, years | 1.04 (1.04, 1.05) | <0.0001 |
| Sex | | |
| Men | Reference | |
| Women | 0.63 (0.59, 0.68) | <0.0001 |
| Race and ethnicity | | |
| Mexican American | Reference | |
| other Hispanic | 0.94 (0.81, 1.08) | 0.3879 |
| non-Hispanic white | 0.80 (0.72, 0.89) | <0.0001 |
| non-Hispanic black | 0.94 (0.83, 1.07) | 0.3444 |
| other races | 0.83 (0.72, 0.95) | 0.0066 |
| Education | | |
| < 9th grade | Reference | |
| 9 –11th grade | 0.67 (0.57, 0.79) | <0.0001 |
| high school graduate | 0.65 (0.56, 0.76) | <0.0001 |
| some college | 0.51 (0.44, 0.59) | <0.0001 |
| college graduate or above | 0.46 (0.39, 0.53) | <0.0001 |
| Marital | | |
| Married/Living with partner | Reference | |
| Widowed/Divorced/Separated | 1.34 (1.22, 1.47) | <0.0001 |
| Never married | 0.54 (0.49, 0.59) | <0.0001 |
| Smoking status | | |
| Never | Reference | |
| Former | 1.64 (1.50, 1.79) | <0.0001 |
| Now | 1.16 (1.06, 1.27) | 0.0013 |
| Alcohol use | | |
| 1–14 | Reference | |
| $\geq$ 15 | 0.92 (0.54, 1.56) | 0.7510 |
| PIR | | |
| $\leq$ 1.3 | Reference | |
| > 1.3 and $\leq$ 3.5 | 1.03 (0.94, 1.12) | 0.5078 |
| > 3.5 | 0.93 (0.85, 1.02) | 0.1112 |
| BMI | | |
| Normal (<25 kg/m$^2$) | Reference | |
| Overweight (25–30 kg/m$^2$) | 1.87 (1.71, 2.04) | <0.0001 |
| Obese ($\geq$30 kg/m$^2$) | 2.79 (2.56, 3.06) | <0.0001 |
| Physical activity | | |
| < 600 | Reference | |
| 600–1500 | 0.84 (0.77, 0.93) | 0.0008 |
| > 1500 | 0.92 (0.84, 1.01) | 0.0901 |
| Daily energy intake (kcal) | 1.00 (1.00, 1.00) | 0.3300 |
| VAI | 1.26 (1.22, 1.29) | <0.0001 |
| LAP | 1.01 (1.01, 1.01) | <0.0001 |
| Hyperlipidemia | | |
| No | Reference | |
| Yes | 2.35 (2.18, 2.54) | <0.0001 |
| Hypertension | | |
| No | Reference | |

(*Continued*)

**Table 2.** (Continued)

| Variables | OR (95%CI) | *P*-value |
|---|---|---|
| Yes | 2.79 (2.58, 3.02) | <0.0001 |
| Cancer | | |
| No | Reference | |
| Yes | 1.86 (1.62, 2.14) | <0.0001 |
| CVD | | |
| No | Reference | |
| Yes | 2.50 (2.15, 2.91) | <0.0001 |

Abbreviations: BMI, body mass index; CI, confidence interval; CVD, cardiovascular diseases; LAP, lipid accumulation product; OR, odds ratio; PIR, poverty income ratio; VAI, visceral obesity index.

## Subgroup analysis

Our results indicated that the positive association between VAI and prediabetes was observed across all subgroups except for those with alcohol use over 15 drinks per week (OR: 1.362, 95% CI: 0.866–2.144), physical activity over 1500 MET-minutes/week (OR: 1.068, 95% CI: 0.995–

**Table 3. Association of VAI and LAP with prediabetes.**

| Exposure | OR (95% CI), *P*-value | | |
|---|---|---|---|
| | Model 1[a] | Model 2[b] | Model 3[c] |
| **VAI** | | | |
| Continuous | 1.294 (1.230, 1.361), <0.0001 | 1.293 (1.226, 1.363), <0.0001 | 1.128 (1.073, 1.185), <0.0001 |
| Q1 (< 0.78) | Reference | Reference | Reference |
| Q2 (0.78–1.27) | 1.248 (1.099, 1.417), 0.0009 | 1.200 (1.052, 1.369), 0.0080 | 0.982 (0.853, 1.130), 0.7974 |
| Q3 (1.27–2.13) | 1.845 (1.626, 2.094), <0.0001 | 1.855 (1.605, 2.144), <0.0001 | 1.259 (1.074, 1.477), 0.0046 |
| Q4 (≥ 2.13) | 2.885 (2.520, 3.302), <0.0001 | 2.912 (2.504, 3.386), <0.0001 | 1.619 (1.354, 1.937), <0.0001 |
| *P* for trend | <0.0001 | <0.0001 | <0.0001 |
| **LAP** | | | |
| Continuous | 1.014 (1.012, 1.016), <0.0001 | 1.012 (1.011, 1.014), <0.0001 | 1.006 (1.004, 1.008), <0.0001 |
| Q1 (< 20.41) | Reference | Reference | Reference |
| Q2 (20.41–36.77) | 2.115 (1.840, 2.431), <0.0001 | 1.711 (1.467, 1.994), <0.0001 | 1.330 (1.137, 1.555), 0.0004 |
| Q3 (36.77–62.28) | 2.933 (2.541, 3.386), <0.0001 | 2.283 (1.963, 2.656), <0.0001 | 1.422 (1.197, 1.690), <0.0001 |
| Q4 (≥ 62.28) | 5.236 (4.499, 6.094), <0.0001 | 4.286 (3.649, 5.036), <0.0001 | 2.164 (1.747, 2.681), <0.0001 |
| *P* for trend | <0.0001 | <0.0001 | <0.0001 |

[a]Model 1: adjusted for no covariates.

[b]Model 2: adjusted for age, gender, race.

[c]Model 3: adjusted for age, gender, race, education, marital, smoking status, BMI, physical activity, hyperlipidemia, hypertension, cancer and CVD.

Abbreviations: BMI, body mass index; CI, confidence interval; CVD, cardiovascular diseases; LAP, lipid accumulation product; OR, odds ratio; VAI, visceral obesity index.

| VAI | OR | 95%CI | | P−value | P for interaction |
|---|---|---|---|---|---|
| **Age** | | | | | 0.001 |
| <50 | 1.265 | (1.170, 1.369) | | <0.001 | |
| >=50 | 1.087 | (1.035, 1.141) | | 0.001 | |
| **Sex** | | | | | 0.005 |
| Men | 1.068 | (1.020, 1.118) | | 0.005 | |
| Women | 1.218 | (1.114, 1.333) | | <0.001 | |
| **Smoking status** | | | | | 0.767 |
| Never | 1.125 | (1.057, 1.199) | | <0.001 | |
| Former | 1.160 | (1.046, 1.286) | | 0.005 | |
| Now | 1.109 | (1.034, 1.189) | | 0.004 | |
| **Alcohol use** | | | | | 0.442 |
| 1−14 | 1.126 | (1.071, 1.183) | | <0.001 | |
| >=15 | 1.362 | (0.866, 2.144) | | 0.181 | |
| **BMI** | | | | | 0.277 |
| Normal (<25 kg/m²) | 1.208 | (1.084, 1.347) | | <0.001 | |
| Overweight (25−30 kg/m²) | 1.100 | (1.030, 1.174) | | 0.004 | |
| Obese (>=30 kg/m²) | 1.134 | (1.070, 1.201) | | <0.001 | |
| **Physical activity** | | | | | 0.065 |
| <600 | 1.196 | (1.119, 1.278) | | <0.001 | |
| 600−1500 | 1.109 | (1.029, 1.196) | | 0.007 | |
| >1500 | 1.068 | (0.995, 1.145) | | 0.067 | |
| **Daily energy intake (kcal)** | | | | | 0.642 |
| <1963 | 1.117 | (1.040, 1.199) | | 0.002 | |
| >=1963 | 1.135 | (1.069, 1.204) | | <0.001 | |
| **Hyperlipidemia** | | | | | 0.103 |
| No | 1.347 | (1.088, 1.668) | | 0.006 | |
| Yes | 1.123 | (1.068, 1.180) | | <0.001 | |
| **Hypertension** | | | | | 0.718 |
| No | 1.121 | (1.060, 1.186) | | <0.001 | |
| Yes | 1.14 | (1.052, 1.235) | | 0.001 | |
| **Cancer** | | | | | 0.175 |
| No | 1.119 | (1.065, 1.176) | | <0.001 | |
| Yes | 1.239 | (1.071, 1.433) | | 0.004 | |
| **CVD** | | | | | 0.815 |
| No | 1.126 | (1.070, 1.186) | | <0.001 | |
| Yes | 1.148 | (0.985, 1.339) | | 0.078 | |

**Fig 2. Subgroup analysis for the association between VAI and prediabetes.** Abbreviation: BMI, body mass index; CI, confidence interval; CVD, cardiovascular diseases; OR, odds ratio; VAI, visceral obesity index.

1.145), and participants with CVD (OR: 1.148, 95% CI: 0.985–1.339). As shown in Fig 2, a significant interaction between age and sex was found; other interaction terms were not significant.

Fig 3 presents a subgroup analysis of the correlation of LAP and prediabetes. The results indicated statistically significant outcomes for all subgroups, including age, gender, smoking status, alcohol use, BMI, physical activity, daily energy intake, hyperlipidemia, hypertension, cancer, and CVD ($P < 0.05$). Meanwhile, except for the interaction terms for age, gender, BMI, and hyperlipidemia, nearly all other interaction terms were not statistically significant.

## Curve fitting and threshold effect analysis

The smooth curve fitting that received the full adjustment demonstrated a non-linear correlation of VAI and prediabetes (S1 Fig). We then conducted a threshold effect analysis and determined that the turning point of the curve was at 4.090 (Table 4).

| LAP | OR | 95%CI | | P−value | P for interaction |
|---|---|---|---|---|---|
| **Age** | | | | | **0.028** |
| <50 | 1.009 | (1.005,1.012) | | <0.001 | |
| >=50 | 1.005 | (1.003,1.007) | | <0.001 | |
| **Sex** | | | | | **0.000** |
| Men | 1.003 | (1.001, 1.006) | | 0.018 | |
| Women | 1.010 | (1.008, 1.013) | | <0.001 | |
| **Smoking status** | | | | | **0.744** |
| Never | 1.006 | (1.004, 1.009) | | <0.001 | |
| Former | 1.007 | (1.003, 1.011) | | <0.001 | |
| Now | 1.005 | (1.002, 1.008) | | 0.001 | |
| **Alcohol use** | | | | | **0.261** |
| 1−14 | 1.006 | (1.004, 1.008) | | <0.001 | |
| >=15 | 1.018 | (1.002, 1.035) | | 0.026 | |
| **BMI** | | | | | **0.020** |
| Normal (<25 kg/m²) | 1.013 | (1.008, 1.019) | | <0.001 | |
| Overweight (25−30 kg/m²) | 1.005 | (1.002, 1.008) | | 0.001 | |
| Obese (>=30 kg/m²) | 1.005 | (1.003, 1.008) | | <0.001 | |
| **Physical activity** | | | | | **0.096** |
| <600 | 1.008 | (1.005, 1.010) | | <0.001 | |
| 600−1500 | 1.006 | (1.003, 1.009) | | 0.001 | |
| >1500 | 1.003 | (1.000, 1.006) | | 0.039 | |
| **Daily energy intake (kcal)** | | | | | **0.591** |
| <1963 | 1.005 | (1.002, 1.008) | | <0.001 | |
| >=1963 | 1.006 | (1.004, 1.008) | | <0.001 | |
| **Hyperlipidemia** | | | | | **0.037** |
| No | 1.012 | (1.007, 1.018) | | <0.001 | |
| Yes | 1.006 | (1.004, 1.008) | | <0.001 | |
| **Hypertension** | | | | | **0.545** |
| No | 1.006 | (1.004, 1.008) | | <0.001 | |
| Yes | 1.005 | (1.002, 1.009) | | 0.005 | |
| **Cancer** | | | | | **0.554** |
| No | 1.006 | (1.004, 1.008) | | <0.001 | |
| Yes | 1.007 | (1.002, 1.012) | | 0.006 | |
| **CVD** | | | | | **0.267** |
| No | 1.006 | (1.004, 1.008) | | <0.001 | |
| Yes | 1.010 | (1.003, 1.016) | | 0.004 | |

**Fig 3. Subgroup analysis for the association between LAP and prediabetes.** Abbreviations: BMI, body mass index; CI, confidence interval; CVD, cardiovascular diseases; LAP, lipid accumulation product; OR, odds ratio.

When VAI was less than 4.090, prediabetes was positively correlated with VAI (OR: 1.238, 95% CI: 1.181–1.298). However, the association was not significant when VAI exceeded 4.090 (OR: 1.031, 95% CI: 0.995–1.068).

Similarly, the correlation of LAP and prediabetes was non-linear (S2 Fig). The inflection point for LAP was at 68.168, as revealed by threshold effect analysis (Table 4). When LAP was below 68.168, a markedly positive correlation between LAP and prediabetes was observed (OR: 1.012, 95% CI: 1.009–1.015). Conversely, when LAP exceeded 68.168, the positive association between LAP and prediabetes still existed, although it was less pronounced (OR: 1.003, 95% CI: 1.002–1.005).

## Discussion

This study is the first to merge and analyze data from the 2007–2020 NHANES to explore the potential correlation of VAI, LAP, and the presence of prediabetes in American adults. The results indicate that both VAI and LAP are closely associated with prediabetes, exhibiting a

**Table 4. Threshold effect analysis of VAI and LAP on prediabetes using piece-wise linear regression[a].**

| Outcome | Adjusted OR (95% CI) | *P*-value |
|---|---|---|
| **Inflection point of VAI** | | |
| < 4.090 | 1.238(1.181, 1.298) | <0.001 |
| > 4.090 | 1.031(0.995, 1.068) | 0.093 |
| Log likelihood ratio test | <0.001 | |
| **Inflection point of LAP** | | |
| < 68.168 | 1.012(1.009, 1.015) | <0.001 |
| > 68.168 | 1.003(1.002, 1.005) | <0.001 |
| Log likelihood ratio test | <0.001 | |

[a]All models were adjusted for: age, gender, race and ethnicity, education, marital, smoking status, BMI, physical activity, hyperlipidemia, hypertension, cancer and CVD.

Abbreviations: BMI, body mass index; CI, confidence interval; CVD, cardiovascular diseases; LAP, lipid accumulation product; OR, odds ratio; VAI, visceral obesity index.

positive correlation. The values of VAI and LAP could potentially be used as predictors of prediabetes.

Individuals with obesity, particularly visceral obesity, have been linked to the development of prediabetes [9, 10]. Techniques used to directly assess visceral fat are expensive and require considerable time to perform; therefore, simple and reliable surrogates of visceral fat are widely used. Although BMI is the most common indicator for detecting obesity, it does not distinguish between fat and muscle or their respective distributions [34]. This limited discriminatory power is particularly relevant for Asians, who tend to exhibit visceral adipose tissue of higher levels compared to Europeans at the identical BMI [35, 36]. Moreover, the loss of muscle, bone mass, and height with age may lead to a reduction in BMI but an increase in fat content. As a result, BMI cut-points are less sensitive to body fatness in older adults than in younger adults, which can lead to misclassification in some older adults [37]. While WC and WHR may be more accurate for measuring abdominal obesity than BMI, they still fail to make a comparison of subcutaneous and visceral adipose tissue. Visceral adipose tissue impacts insulin metabolism because it releases more free fatty acids than subcutaneous fat [38, 39].

VAI combines anthropometric and metabolic measurements and has been reported to correlate highly with visceral adipose tissue as evaluated by MRI [40]. The LAP is an effective tool for indicating the combined anatomic and physiologic changes caused by the deposition of visceral fat [18]. Gu et al. [41] and Liu et al. [42] have indicated that VAI showed a positive correlation to prediabetes in Chinese adults. However, another Chinese study found no significant association [43]. This discrepancy might be due to the fact that VAI was developed for Caucasians and body fat distribution varies among ethnicities [44]; thus, VAI may not be suitable for the Chinese population. Song et al. [45] documented that LAP was positively correlated with impaired fasting glucose in Chinese people, especially in females. Since this study was limited to the Chinese middle-aged and elderly population, its applicability to younger populations and other ethnicities may be limited due to age and ethnicity factors [44, 46]. A German study [25] showed that VAI and LAP are useful indices for discriminating against prediabetes/diabetes in males and females, although it did not assess associations with VAI and LAP and prediabetes separately. Previous studies have established that VAI and LAP were superior to traditional anthropometric measurements (i.e., BMI, WHR, and WC) for prediabetes risk prediction [23, 24]. In another study, Nusrianto et al. [47] demonstrated that elevated LAP and VAI were correlated with a worsening glycemic status; VAI was associated with a risk of

prediabetes in both sexes, while LAP was associated in women but not in men. These inconsistent results might be due to differences in sample size, race, region, and study design. Our study highlighted the strong association of VAI and LAP with prediabetes, consistent with previous findings. Therefore, in order to reduce the negative health outcomes of visceral obesity, individuals must take the initiative to change their unhealthy lifestyle behaviors, including eating habits, and employ other methods.

Our study demonstrated that the relationships between VAI or LAP and the risk of prediabetes were non-linear, with inflection points at 4.090 and 68.168, respectively. VAI was positively correlated with prediabetes when it was below 4.090; however, the risk of prediabetes remained essentially stable when VAI exceeded 4.090, though not significantly. The positive association between LAP and prediabetes persisted whether LAP was below or above 68.168. Qin et al. [48] found revealed a non-linear positive correlation between VAI and fasting plasma glucose levels, with an inflection point at 4.02. For individuals with VAI below 4.02, the FPG level increased rapidly with rising VAI (β: 0.73, 95% CI: 0.59–0.87). For those with VAI above 4.02, FPG displayed a relatively mild upward trend (β: 0.23, 95% CI: 0.07–0.40). In the study of Song et al. [45], it was observed that individuals with LAP values in the lowest quartile faced a markedly elevated risk of impaired fasting glucose when contrasted to individuals in the highest quartile.

Insulin resistance is a hallmark of prediabetes, which is linked to excessive visceral fat [49, 50]. VAI and LAP have been suggested as preliminary indicators that could signal the presence of insulin resistance [51–53]. The potential mechanisms by which VAI and LAP influence the outcome of prediabetes may include: Visceral fat exhibits high lipolytic activity, leading to an increased free fatty acid load in the portal circulation, which promotes hepatic fat accumulation and insulin resistance [35, 36]. Visceral adipocytes produce and release a series of adipokines, including interleukin-6, adiponectin, and leptin, which may lead to increased insulin resistance [54, 55]. An excess of visceral adipose tissue activates macrophages secreting considerable inflammatory cytokines, resulting in diminished insulin sensitivity [56]. Adiponectin, an adipokine mainly secreted by adipocytes, can regulate glucose and lipid metabolism. Elevated visceral adipose tissue decreases adiponectin levels [57], potentially exacerbating insulin resistance [58].

Our research exhibits some advantages. Initially, we analyzed data drawn from a broad and demographically diverse sample spanning the entire nation, allowing the weighted results to be interpreted as reflective of the U.S. population. Second, we employed an advanced statistical method (multiple imputation) to address missing data, reducing potential bias and enhancing the statistical power of our results. Additionally, the study's findings were stable and robust in subgroup analyses. Despite these strengths, we must acknowledge certain limitations. First, NHANES is a cross-sectional study; thus, the increases in VAI and LAP could be a consequence of elevated blood glucose or insulin resistance occurring after the onset of prediabetes, rather than the cause of prediabetes. Longitudinal studies will be necessary to validate the utility of VAI and LAP as predictive markers for prediabetes. Secondly, although we adjusted for known potential confounders, other residual confounding factors might remain, which could influence the associations. Lastly, our research merely covered American participants, and the findings might not extend to other countries and populations.

## Conclusions

Drawing from a nationally representative population, the findings from this investigation indicate that VAI or LAP with higher levels may be linked to a greater likelihood of developing prediabetes. This insight is of considerable importance to public health, advocating for the use

of VAI or LAP as simple and practical indicators for clinical assessment of prediabetes risk. To solidify the understanding of how these associations work and to verify their causative nature, further extensive, well-conducted research studies are required.

## Supporting information

**S1 Fig. Smooth curve fitting of VAI and prediabetes.**
(TIF)

**S2 Fig. Smooth curve fitting of LAP and prediabetes.**
(TIF)

## Acknowledgments

We thank the National Center for Health Statistics (NCHS) for providing the NHANES data used in this study. We also acknowledge the participants and staff of NHANES, and our colleagues for their support and feedback during the preparation of this manuscript.

## Author Contributions

**Data curation:** Li-Ting Qiu, Ji-Dong Zhang.

**Formal analysis:** Ji-Dong Zhang.

**Methodology:** Li-Ting Qiu, Bo-Yan Fan, Ling Li.

**Software:** Li-Ting Qiu, Bo-Yan Fan.

**Supervision:** Gui-Xiang Sun.

**Writing – original draft:** Li-Ting Qiu, Ling Li, Gui-Xiang Sun.

**Writing – review & editing:** Gui-Xiang Sun.

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
