## [Decision Letter · Decision Letter 0]

15 Aug 2024

PONE-D-24-10840Association of Visceral Adiposity Index and Lipid Accumulation Products with Prediabetes in US Adults from NHANES 2007-2020: A Cross-sectional StudyPLOS ONE

Dear Dr. Sun,

Thank you for submitting your manuscript to PLOS ONE. After careful consideration, we feel that it has merit but does not fully meet PLOS ONE’s publication criteria as it currently stands. Therefore, we invite you to submit a revised version of the manuscript that addresses the points raised during the review process.

We look forward to receiving your revised manuscript.

Kind regards,

Demitri Constantinou, MD

Academic Editor

PLOS ONE

“This study was supportded by the National Natural Science Foundation of China (81973670), Central Subsidized Chinese Medicine Special Funds - National Medical Master Sun Guangrong Hunan Workshop Project (2015).”

4. Please note that your Data Availability Statement is currently missing the repository name and/or the DOI/accession number of each dataset OR a direct link to access each database. If your manuscript is accepted for publication, you will be asked to provide these details on a very short timeline. We therefore suggest that you provide this information now, though we will not hold up the peer review process if you are unable.

5. We notice that your supplementary figures are uploaded with the file type 'Figure'. Please amend the file type to 'Supporting Information'. Please ensure that each Supporting Information file has a legend listed in the manuscript after the references list.

Reviewers' comments:

Reviewer's Responses to Questions

**Comments to the Author**

1. Is the manuscript technically sound, and do the data support the conclusions?

Reviewer #1: Yes

Reviewer #2: Yes

2. Has the statistical analysis been performed appropriately and rigorously? 

Reviewer #1: Yes

Reviewer #2: Yes

3. Have the authors made all data underlying the findings in their manuscript fully available?

Reviewer #1: Yes

Reviewer #2: No

4. Is the manuscript presented in an intelligible fashion and written in standard English?

Reviewer #1: Yes

Reviewer #2: Yes

5. Review Comments to the Author

Reviewer #1: This cross-sectional study evaluated the association of the visceral adiposity index (VAI) and the lipid accumulation product (LAP) with prediabetes in 12,564 US adults from NHANES 2007-2020. VAI and LAP values were found to be positively associated with the presence of prediabetes. Based on their findings, the authors conclude that the VAI and LAP indices may be used as predictors of prediabetes. The topic is important and, generally, the manuscript is very well written. However, the manuscript could benefit from revisions to address the following concerns:

1. The authors appear to use the words "prevalence" and "incidence" as if they are interchangeable. For example, in 3.2, the authors use the term "incidence rates" when, given the cross-sectional as opposed to longitudinal study design, "prevalence" is the correct term. On a similar note, in the discussion, the statement that "This study is the first to merge and analyze data from the 2007–2020 NHANES to explore the potential correlation of VAI, LAP, and the incidence of developing prediabetes in American adults." should be changed to something like "This study is the first to merge and analyze data from the 2007–2020 NHANES to explore the potential correlation of VAI, LAP, and the presence of prediabetes in American adults." In the abstract, "To evaluate the connection between VAI or LAP and the incidence of prediabetes" should be changed to something like "To evaluate the connection between VAI or LAP and the presence of prediabetes".

2. Was information available on a family history of diabetes and gestational diabetes in women? If so, the authors should comment on why such data were not considered in their analyses. Also, if feasible, it would be of interest to know how the VAI and LAP compare to the ADA/CDC Prediabetes Risk Test

(https://nationaldppcsc.cdc.gov/s/article/New-American-Diabetes-Association-ADA-CDC-Prediabetes-Risk-Test).

3. When referring to METs in connection with physical activity volume, the authors should use the terminology "MET-minutes/week."

4. When discussing limitations of the study, the authors state that "First, NHANES is a cross-sectional study; thus, we can establish correlations but not causal relationships between prediabetes and VAI and LAP." That statement is correct but the authors should further elaborate by discussing the following even more relevant potential limitation: Given the cross-sectional nature of the study, it is conceivable that neither the VAI or LAP are actually strongly predictive of the future development of type 2 diabetes (or, predictive to a lesser degree than appears evident in this study) because it is possible that it is only once a person actually develops prediabetes that the elevated blood glucose and/or insulin resistance cause an increase in triglycerides and decrease in HDL (and, thus, an increased VAI and LAP). Prediabetes could be predictive of a future increase in VAI and LAP rather than an increase in VAI and LAP being predictive of the future development of prediabetes.

Reviewer #2: This research aims at two novel indices and its association with Prediabetes in US adults.

Introduction: The introduction does not include the research conducted with VAI and LAP on ethnicities other than South Asian population. The objective is stated but is deficient by hypothesis.

Methodology:

The study design is not mentioned and the sample size estimation using multistage probability sampling requires description. I understand the data is collected by NHANES. How are the anthropometric measurements recorded and We assume they follow standard guidelines but this requires a mention.

The covariates data such as social, demographics, caloric intake, physical activity, smoking and alcohol history, were they all available in the database or these information were later recorded from participants. This can make a difference because the biochemical and body composition measurements must match with the timing when history is collected from participants.

Statistical analysis:

The regression analysis adopted requires explanation stepwise. Was the multivariate analysis conducted based on the results from univariate analysis? Three models were adopted for multivariate analysis- this is mentioned later in the results but not in methodology. Why is the OR mentioned only once for three models of multivariate analysis (line 226 & 228). The table show the OR for all three models.

Table 3, first column, What is the meaning of reference? This requires description.

Table 4 mentions inflection point, but the determination of inflection point needs to be mentioned.

6. PLOS authors have the option to publish the peer review history of their article (what does this mean?). If published, this will include your full peer review and any attached files.

Reviewer #1: No

Reviewer #2: No

---

## [Author Response · Author response to Decision Letter 0]

29 Aug 2024

Thank you for your letter and for the reviewers’ comments concerning our manuscript entitled “Association of Visceral Adiposity Index and Lipid Accumulation Products with Prediabetes in US Adults from NHANES 2007-2020: A Cross-sectional Study” (Manuscript ID: PONE-D-24-10840). Those comments are all valuable and very helpful for revising and improving our paper, as well as the important guiding significance to our researches. We have studied comments carefully and have made correction which we hope meet with approval. Revised portion are marked in highlight yellow (normal revision) in the paper. The main corrections in the paper and the responds to the reviewer’s comments are as following:

Reviewer # 1

This cross-sectional study evaluated the association of the visceral adiposity index (VAI) and the lipid accumulation product (LAP) with prediabetes in 12,564 US adults from NHANES 2007-2020. VAI and LAP values were found to be positively associated with the presence of prediabetes. Based on their findings, the authors conclude that the VAI and LAP indices may be used as predictors of prediabetes. The topic is important and, generally, the manuscript is very well written. However, the manuscript could benefit from revisions to address the following concerns.

Response: Thank you for your letter and your comments. Those comments are all valuable and very helpful for revising and improving our paper. Our study used data from the National Health and Nutrition Examination Survey from 2007 to 2020, ultimately enrolling and analysing 12,564 participants; and concluded that VAI and LAP levels were positively related to an increased prevalence of prediabetes in American adults. These data suggest that VAI and LAP can be used as a new anthropometric indicator for predicting prediabetes, providing clues for population screening and early clinical disease intervention. Our study has some limitations, including a cross-sectional design that limits causal inference, the potential impact of residual confounding, national differences, etc. If there are any other modifications we could make, we would like very much to modify them and we really appreciate your help.

The authors appear to use the words "prevalence" and "incidence" as if they are interchangeable. For example, in 3.2, the authors use the term "incidence rates" when, given the cross-sectional as opposed to longitudinal study design, "prevalence" is the correct term. On a similar note, in the discussion, the statement that "This study is the first to merge and analyze data from the 2007–2020 NHANES to explore the potential correlation of VAI, LAP, and the incidence of developing prediabetes in American adults." should be changed to something like "This study is the first to merge and analyze data from the 2007–2020 NHANES to explore the potential correlation of VAI, LAP, and the presence of prediabetes in American adults." In the abstract, "To evaluate the connection between VAI or LAP and the incidence of prediabetes" should be changed to something like "To evaluate the connection between VAI or LAP and the presence of prediabetes".

Response: Thanks for this positive comment. We are very sorry for the incorrect use of the word. Based on your suggestion, we’ve changed “incidence” to “prevalence” (See Lines26, Page.1; Lines211, Page.9; Lines302, Page.15). Furthermore, in order to avoid similar problems, we have also carried out a thorough examination of the full-text.

Was information available on a family history of diabetes and gestational diabetes in women? If so, the authors should comment on why such data were not considered in their analyses. Also, if feasible, it would be of interest to know how the VAI and LAP compare to the ADA/CDC Prediabetes Risk Test

(https://nationaldppcsc.cdc.gov/s/article/New-American-Diabetes-Association-ADA-CDC-Prediabetes-Risk-Test).

Response: Thank you for your thorough review and valuable feedback. Regarding the first question, the NHANES database does indeed contain information on family history of diabetes and gestational diabetes. However, due to a significant amount of missing data in these questionnaire items, including this information in our analysis could potentially bias the results. Therefore, we did not incorporate family history of diabetes or gestational diabetes into our analytical model.

Additionally, due to the extensive missing data on gestational diabetes and family history of diabetes in the NHANES database, we were unable to calculate the ADA/CDC Prediabetes Risk Test scores. Consequently, feel sincerely sorry that we could not directly compare LAP and VAI with the ADA/CDC Prediabetes Risk Test. 

When referring to METs in connection with physical activity volume, the authors should use the terminology "MET-minutes/week."

Response: We were really sorry for our careless mistakes. Thank you for your reminder. The “METs” has been corrected on “MET-minutes/week”(See Lines154, Page.6; See Lines195, Page.8; See Lines224-226, Page.10). At the same time, we have carefully reviewed the manuscript to avoid the same mistakes.

When discussing limitations of the study, the authors state that "First, NHANES is a cross-sectional study; thus, we can establish correlations but not causal relationships between prediabetes and VAI and LAP." That statement is correct but the authors should further elaborate by discussing the following even more relevant potential limitation: Given the cross-sectional nature of the study, it is conceivable that neither the VAI or LAP are actually strongly predictive of the future development of type 2 diabetes (or, predictive to a lesser degree than appears evident in this study) because it is possible that it is only once a person actually develops prediabetes that the elevated blood glucose and/or insulin resistance cause an increase in triglycerides and decrease in HDL (and, thus, an increased VAI and LAP). Prediabetes could be predictive of a future increase in VAI and LAP rather than an increase in VAI and LAP being predictive of the future development of prediabetes.

Response: Thank you for your insightful suggestions on improving the discussion of the study’s limitations. We completely agree with your observation that, as a cross-sectional study, our research is inherently limited in its ability to determine the true predictive value of VAI and LAP for the future development of type 2 diabetes. Indeed, as you suggested, the observed elevations in these indices may result from metabolic changes associated with prediabetes, rather than being precursors to it.

To address this, we will expand the discussion in the revised manuscript to include this important point: NHANES is a cross-sectional study; thus, the increases in VAI and LAP could be a consequence of elevated blood glucose or insulin resistance occurring after the onset of prediabetes, rather than the cause of prediabetes. Longitudinal studies will be necessary to validate the utility of VAI and LAP as predictive markers for prediabetes (See Lines379-383, Page.18).

Reviewer # 2

This research aims at two novel indices and its association with Prediabetes in US adults.

Response: Thanks very much for taking your time to review this manuscript. We really appreciate all your comments and suggestions. Our study utilized data from the National Health and Nutrition Examination Survey spanning from 2007 to 2020, ultimately enrolling and analyzing 12,564 participants. It was concluded that VAI and LAP levels were positively associated with an elevated prevalence of prediabetes among American adults. These data imply that VAI and LAP can serve as a novel anthropometric indicator for predicting prediabetes, offering cues for population screening and early clinical disease intervention. However, our study has certain limitations, such as a cross-sectional design which restricts causal inference, the potential influence of residual confounding, and national disparities, among others. Based on your comments and suggestions, we have made careful modifications to the original manuscript, and carefully proof-read the manuscript. We would like to thank the referee again for taking the time to review our manuscript.

Comments:

The introduction does not include the research conducted with VAI and LAP on ethnicities other than South Asian population. The objective is stated but is deficient by hypothesis.

Response: Thank you for your valuable feedback. We have revised the Introduction section to include references to studies conducted on populations other than South Asians that have utilized the Visceral Adiposity Index (VAI) and Lipid Accumulation Product (LAP) (See Lines92-94, Page.4). These studies demonstrate that while the cutoff points for VAI and LAP may vary due to ethnic differences, their effectiveness in predicting prediabetes remains significant across diverse populations. To further enhance the rigor of our study, we have explicitly stated our hypothesis in the revised Introduction: "The hypothesize of this study is that lower levels of VAI or LAP are associated with a significantly lower risk of prediabetes." (See Lines101-102, Page.4) This hypothesis aims to provide a theoretical foundation for the diversity and broad applicability of our research.

Methodology:

The study design is not mentioned and the sample size estimation using multistage probability sampling requires description. I understand the data is collected by NHANES. How are the anthropometric measurements recorded and We assume they follow standard guidelines but this requires a mention.

Response: Thank you for highlighting this important aspect of our study. In the revised Methods section, we have provided a detailed description of the study design. Our study is based on data from the National Health and Nutrition Examination Survey (NHANES) and employs a cross-sectional study design. NHANES utilizes a complex, multistage probability sampling method to select a sample that is representative of the non-institutionalized population of the United States. Specifically, NHANES first selects primary sampling units (PSUs) across the country, followed by stratified random sampling within each PSU to choose households, and finally, individuals are randomly selected from eligible household members.

Regarding sample size estimation. Although NHANES itself does not provide explicit sample size estimations for individual studies, it surveys approximately 5,000 individuals annually. The use of sample weights provided by NHANES allows for unbiased national estimates, a process we have described in our statistical analysis section. Specifically, the variables SDMVSTRA and SDMVPSU were utilized to ensure accurate national estimates due to the complex survey design employed in NHANES. We have now elaborated on this sampling process and its implications for the study's conclusions in the revised manuscript (See Lines111-116, Page.4-5).

The covariates data such as social, demographics, caloric intake, physical activity, smoking and alcohol history, were they all available in the database or these information were later recorded from participants. This can make a difference because the biochemical and body composition measurements must match with the timing when history is collected from participants.

Response: Thank you for your insightful question regarding the covariate data. We have clarified this in the revised manuscript (See Lines116-120, Page.4-5). All anthropometric measurements, such as height, weight, and waist circumference, were recorded following NHANES' standardized protocols, ensuring accuracy and consistency. Regarding the covariates, including social and demographic information, caloric intake, physical activity, smoking, and alcohol history, these data were also sourced directly from the NHANES database. These variables were collected concurrently with the biochemical and body composition measurements during the participants' medical examinations at NHANES' Mobile Examination Centers (MECs). This synchronization ensures that the timing of the data collection is aligned, thereby maintaining the temporal consistency between the covariates and the primary variables in our study. As a result, there is no risk of temporal mismatch, which could otherwise compromise the validity of our findings.

Statistical analysis:

The regression analysis adopted requires explanation stepwise. Was the multivariate analysis conducted based on the results from univariate analysis? Three models were adopted for multivariate analysis- this is mentioned later in the results but not in methodology. Why is the OR mentioned only once for three models of multivariate analysis (line 226 & 228). The table show the OR for all three models.

Response: We sincerely appreciate your insightful suggestions regarding the statistical analysis. To address your concerns, we have provided a more detailed explanation of the univariate and multivariate analyses in the revised manuscript.

In our study, the univariate analysis was conducted primarily to assess the strength and direction of the association between each individual covariate and the risk of prediabetes. Based on these initial findings, we developed three multivariate logistic regression models to further explore the relationship between the Visceral Adiposity Index (VAI), Lipid Accumulation Product (LAP), and prediabetes.

The Model 1, was constructed without any adjustments, serving as the baseline analysis. In Model 2, adjustments were made for key demographic variables, specifically gender, age, and race, to account for their potential confounding effects. Finally, Model 3 included a comprehensive set of covariates, which were retained in the model if their inclusion resulted in a change of more than 10% in the effect estimate. This stepwise approach allowed us to systematically evaluate the impact of these covariates on the relationship between VAI, LAP, and prediabetes.

We have now included these detailed explanations in the Methods section of the revised manuscript (See Lines174-179, Page.7). Furthermore, in response to your suggestion, we have clarified the presentation of OR for each of the three models in the Results section, ensuring that our analysis is both transparent and robust.

Table 3, first column, What is the meaning of reference? This requires description.

Response: Thank you for your attention to the clarity of Table 3. In the revised manuscript, we have added an explanation (See Lines236-237, Page.11) to clarify the meaning of "reference" in the first column of Table 3. Specifically, the term "reference" refers to the control group, which typically represents the lowest quartile (Q1) or the group not exposed to a particular risk factor. This group serves as the baseline for comparison, allowing us to evaluate the OR of the other quartiles or categories relative to this reference group.

Table 4 mentions inflection point, but the determination of inflection point needs to be mentioned.

Response: Thank you for your insightful comment regarding the determination of the inflection point mentioned in Table 4. In the revised manuscript, we have provided a detailed explanation of how the inflection point was identified. The inflection point was determined through a threshold effect analysis, which allows us to detect nonlinear relationships between the Visceral Adiposity Index (VAI), Lipid Accumulation Product (LAP), and the risk of prediabetes. Initially, we conducted a smooth curve fitting analysis to explore the potential nonlinear relationship between VAI, LAP, and prediabetes. Following this, we applied threshold effect analysis to pinpoint the specific values where the relationship between VAI and LAP with prediabetes risk exhibited significant changes. In this study, the inflection point for VAI was identified at 4.090, and for LAP, it was identified at 68.168. These details have been thoroughly described in the Methods section (See Lines182-185, Page.7).

We acknowledge the reviewer’s comments and suggestions very much, which are valuable in improving the quality of our manuscript. Thank you and all the reviewers for the kind advice.

Sincerely yours.

---

## [Editor Report · Decision Letter 1]

10 Sep 2024

Association of Visceral Adiposity Index and Lipid Accumulation Products with Prediabetes in US Adults from NHANES 2007-2020: A Cross-sectional Study

PONE-D-24-10840R1

Dear Dr. Sun,

We’re pleased to inform you that your manuscript has been judged scientifically suitable for publication and will be formally accepted for publication once it meets all outstanding technical requirements.

Kind regards,

Demitri Constantinou, MD

Academic Editor

PLOS ONE
---

## [Editor Report · Acceptance letter]

20 Sep 2024

PONE-D-24-10840R1 

PLOS ONE

Dear Dr. Sun, 

I'm pleased to inform you that your manuscript has been deemed suitable for publication in PLOS ONE. Congratulations! Your manuscript is now being handed over to our production team.

Kind regards, 

on behalf of

Professor Demitri Constantinou 

Academic Editor

PLOS ONE